# Establishing the Link between X-Chromosome Aberrations and *TP53* Status, with Breast Cancer Patient Outcomes

**DOI:** 10.3390/cells12182245

**Published:** 2023-09-11

**Authors:** Franco Caramia, Terence P. Speed, Hui Shen, Ygal Haupt, Sue Haupt

**Affiliations:** 1Peter MacCallum Cancer Centre, Melbourne, VIC 3000, Australia; franco.caramia@petermac.org (F.C.); ygal.haupt@petermac.org (Y.H.); 2Sir Peter MacCallum Department of Oncology, The University of Melbourne, Melbourne, VIC 3010, Australia; 3Walter and Eliza Hall Institute for Medical Research, Parkville, VIC 3052, Australia; terry@wehi.edu.au; 4Van Andel Institute, Grand Rapids, MI 49503, USA; hui.shen@vai.org

**Keywords:** X chromosome, XCI, breast cancer, TP53, TCGA, METABRIC

## Abstract

Ubiquitous to normal female human somatic cells, X-chromosome inactivation (XCI) tightly regulates the transcriptional silencing of a single X chromosome from each pair. Some genes escape XCI, including crucial tumour suppressors. Cancer susceptibility can be influenced by the variability in the genes that escape XCI. The mechanisms of XCI dysregulation remain poorly understood in complex diseases, including cancer. Using publicly available breast cancer next-generation sequencing data, we show that the status of the major tumour suppressor *TP53* from Chromosome 17 is highly associated with the genomic integrity of the inactive X (Xi) and the active X (Xa) chromosomes. Our quantification of XCI and XCI escape demonstrates that aberrant XCI is linked to poor survival. We derived prognostic gene expression signatures associated with either large deletions of Xi; large amplifications of Xa; or abnormal X-methylation. Our findings expose a novel insight into female cancer risks, beyond those associated with the standard molecular subtypes.

## 1. Introduction

The X chromosome is a major protagonist in the study of cancers, as it encodes tumour suppressors, oncogenes and regulators of immunity, hormones, and reproduction ([1] and references within). One copy of two X chromosomes is transcriptionally silenced in female mammals, which balances the number of active X chromosomes between the sexes [2,3,4]. This dosage compensation mechanism, known as X-chromosome inactivation (XCI), is initiated embryonically by the *X-inactive-specific transcript* (*XIST*) and may continue to support the inactivation of certain genes in some adult cell types [5]. More commonly, XCI appears to be maintained by DNA methylation in gene promoter regions [6]. A genome-wide DNA methylation study highlighted that X-chromosome dosage compensation via gene promoter methylation may adapt in the presence of copy number aberrations (CNAs) in certain breast cancers [2]. XCI does not silence the whole Xi, however, with 10–15% of genes escaping. Escaping XCI is a known and well-studied phenomenon [3], but has proven challenging to characterise completely, as it is highly variable between tissue types, cell populations, and individuals. Several studies have drawn maps of XCI escaping [4,7], and, recently, age-acquired XCI skewness has been associated with adverse health outcomes [8]. However, the breadth of functional consequences imposed by variable XCI escape remains poorly understood. Strictly in conjunction with XCI, an intriguing moderation of the expression of the Xa occurs in a process termed X-chromosome upregulation (XCU). This is separate from XCI escape and is not uniform across the X genes [9]. The emergent function of XCU in females is to ensure that, overall, X genes are expressed at levels comparable to autosome genes and to the hyperactivated male X [10]. Importantly, with the implications of the X chromosome in cancer defence, we hypothesise that X-chromosome alterations may dictate the course of cancer emergence.

The complexities of XCI have impeded our understanding of how the X chromosome protects females from cancers, despite decades of study. Next-generation sequencing (NGS) and the availability of large cancer cohorts are now enabling the discovery of events underlying malignant phenotypes. One landmark study demonstrated six tumour-suppressor genes that escape XCI and may contribute to cancer sex-disparity (ATRX, CNKSR2, DDX3X, KDM5C, KDM6A, and MAGEC3). Higher expression levels of these genes may offer females greater cancer protection than males [11]. Pertinently, the poor prognosis in ovarian cancers has been associated with XCI dysregulation, involving excessive or irregular escaping from XCI, to an extent that ranged from partial to complete [12,13]. These contrasting scenarios highlight the need for in-depth analyses to grasp the influence of normal and altered XCI in cancer progression, in specific tissues.

Breast carcinomas (BRCAs) are standardly grouped into molecular subtypes, with established differences in tumour progression, immune landscape, treatments, and survival [14,15]. Notably, subtypes also differ in the extent of X-chromosome damage, with the Basal subtype reported to be the most impacted [16,17,18] and to display the greatest misbehaviour in *XIST* regulation [19]. In addition, Basal BRCAs are the heaviest carriers of somatic mutations for the tumour-suppressor protein p53, encoded in gene *TP53* [20]. P53 is a transcription factor, described as the “guardian of the genome”, for its protective cellular responses to stress [21]. DNA integrity is maintained by both transcription-dependent and independent functions of the p53 function [22,23]. p53 actively resists the accumulation of abnormal chromosomal numbers, which consequently averts the major cancer risks associated with aneuploidy [24]. Nefariously, when *TP53* is mutated, the acquisition of aneuploidy is a risk for the enhancement of its new oncogenic functions [25,26].

Specific to the X chromosome, we previously showed that the loss of p53 has been associated with aberrant XCI in female mice during development [27]. In sporadic cancers, we demonstrated that males are particularly vulnerable to alterations of the X chromosome’s p53 network [28]. Together, these findings reinforce links between p53 integrity and the fidelity of the X chromosome.

Hypothesising that p53 status in BRCAs influences *XIST*, and XCI and its aberrations, we address this link for the first time, through systematically interrogating NGS data: BRCA from The Cancer Genome Atlas (TCGA-BRCA) and the Molecular Taxonomy of Breast Cancer International Consortium (METABRIC). We use X-chromosome methylation to quantify XCI and explore the ramifications of large deletions of the Xi and large Xa amplifications. Adopting our unique method to assess somatic mutation silencing in the X chromosome [28], we show that the most aggressive BRCAs express an elevated number of somatic mutations in X-linked genes in conjunction with overrepresented *TP53* mutations. Based on our data, we report an association between high and low levels of XCI and worse survival outcomes, and we establish clinically prognostic X-chromosome signatures that offer insight beyond the standard BRCA subtyping, with therapeutic ramifications.

## 2. Materials and Methods

### 2.1. Data Acquisition

Publicly available NGS data from TCGA, including quantified RNA-seq counts, methylation beta-values, allele-specific copy number segments, and annotated somatic mutations, were accessed using the TCGAbiolinks package in R [29]. Somatic mutations were filtered to exclude those annotated as ‘Silent’. Previously called BRCA, ATAC-seq peaks were also publicly available [30]. Permission to access controlled alignment RNA-seq files (bam) was obtained through the database of genotypes and phenotypes (dbGaP), and processing was carried out using the Seven Bridges CGC platform. PAM50 subtypes for TCGA-BRCA were used as published in [31].

For the METABRIC dataset, normalised microarray gene expression, clinical data, and PAM50 subtypes were accessed through CBioPortal (https://www.cbioportal.org, accessed on 29 June 2020). Promoter methylation estimates from reduced representation bisulfite sequencing were used as published in [2].

### 2.2. Clustering Analysis

Ward’s hierarchical clustering method was used to classify samples by allele-specific copy number segments and methylation beta-values in R using the daisy method [32] to build the dissimilarity matrix.

### 2.3. Gene Promoter Methylation

Gene methylation beta-values were annotated according to probe’s proximity to the gene transcription starting site (TSS). We used the mean beta-value of probes located within 200 base pairs (bp) of the TSS. If no probes were located within 200 bp, we used the probes located within 1500 bp; otherwise, no methylation beta-value was assigned to the gene.

### 2.4. Somatic Mutation Expression

RNA-mutated allele frequency (RMAF) from somatic mutations located in the X chromosome were calculated using RMAFster (https://github.com/fcaramia/RMAFster). Somatic mutations were classified as ‘expressed’, ‘non.expressed’, and ‘escape’ according to RMAFs (RMAF ≥ 0.75, RMAF ≤ 0.25, and 0.25 < RMAF < 0.75, respectively).

### 2.5. Differential Expression Analysis

We built four gene expression signatures using differentially expressed genes analysed using the limma library in R [33]. We accounted for biases introduced by uneven numbers of BRCA molecular subtypes in the comparison groups by including the PAM50 classification as a covariate in the model matrix. Genes were filtered by false discovery rate (*FDR* ≤ 0.05) and log fold change (*logFC* > 0 or *logFC* < 0).

### 2.6. Survival Analysis

We assessed the prognostic power of four different gene expression signatures using the METABRIC dataset. Gene signatures were compiled using the ‘sig.score’ function in the R package genefu [34]. Kaplan–Meier survival curves were generated by partitioning cases by the median value of the respective signature score. Hazard ratios were derived using Cox proportional hazard survival models using the respective signature score with endpoints of disease-free survival of up to 20 years. All plots and tests were generated in R.

### 2.7. Statistical Tests

All hypothesis testing was performed in R version 4.2.0. K–W: Kruskal–Wallis test.

## 3. Results

### 3.1. Aggressive Breast Carcinomas with TP53 Mutations Contain Structural X-Chromosome Aberrations

Regulated XCI maintenance in adult females has largely been attributed to methylation of Xi gene promoters [8], with recent findings exposing that *XIST* may also influence XCI, in at least some types of adult female cells [35]. Initially, we speculated that the levels of disruption of X gene promoter methylation may predict the ensuing aggressiveness in BRCA. To test this hypothesis, we conducted an extensive study of the X chromosome using NGS.

An analysis of the survival data from METABRIC for up to five years showed that the most frequent subtypes, ranked from least to most aggressive, were: Luminal A, Luminal B, Her2, and Basal BRCAs (shown by us in Appendix A; p≈0, log-rank test; and, similarly, by others [36]). Consistent with our speculations, we observed altered promoter methylation at diagnosis between BRCA subtypes (p≈0; K–W test; Figure 1a), but without an absolute alignment with disease aggression, implying additional influences. This supported our next approach, which was to examine *XIST* expression in the biopsied samples, and, significantly, relative to the normal breast, we found progressively decreasing levels across Luminal A and B, Her2, and Basal BRCA, respectively (p≈0; K–W test; Figure 1b). We previously reported that the presence of wild-type *TP53* (WT-*TP53*) positively regulated *XIST* expression in normal mouse development [27]. The ramifications of altered *TP53* status for X promoter methylation and *XIST* expression in adult sporadic cancers have not been reported. Accordingly, we tested these differences in adult BRCAs. We show that X promoter methylation and *XIST* levels are significantly lower in adult BRCAs that have acquired *TP53* somatic mutations (Mt-*TP53*) (p≈0, respectively; Wilcoxon test; Figure 1c,d). We further predicted that reduced X gene promoter methylation levels and reduced *XIST* expression may be signalling deep Xi deletions. Of clinical relevance, we identified lower levels of X promoter methylation and *XIST* expression in BRCA, to correlate with shorter disease-specific survival in METABRIC (p=0.06 and p=0.006, respectively; log-rank test; Figure 1e,f). Vitally, our data predict that the presence of Wt-*TP53* is relevant to proper XCI and *XIST* expression in adult human tissues, just as it proved to be in mouse development [27], although the mechanisms may differ. These observations prompted us to explore the association between BRCA aggressiveness, *TP53* status, and Xi abnormalities beyond the scope of previous studies [8,16,18] and recent epigenetic analyses [4].

To investigate the prevalence of Xi abnormalities, we performed hierarchical clustering using the TCGA-BRCAs of the four most frequent subtypes, based on large Xi deletions (Figure 2a; Appendix A), and identified two groups of samples: tumours showing a complete or extended deletion of Xi (Xi-large-deletion), and tumours with a mostly unaltered Xi (Xi-unaltered). A disproportionately high number of the three most aggressive subtypes exhibited large or complete Xi deletions—Basal (60%; 111/185), LumB (36%; 74/205), and Her2 (35%; 28/80), respectively—as compared to the least aggressive subtype Luminal A (17%; (92/545); p≈0; ChiSq test; Figure 2b). Strikingly, we found that Mt-*TP53* samples displayed a higher proportion of large Xi deletions (51%; 185/363) compared to Wt-*TP53* samples (18%; 120/652). We reasoned that, if an X chromosome is deleted from the pair, XCI becomes futile. Aligning with this, we observed reduced levels of *XIST* and X promoter methylation in the Xi-large deletion group (p≈0; Wilcoxon test; Figure 2a,b). These lower levels are also observed in samples with *TP53* mutations, as shown in Figure 1c,d. To further explore the involvement of *TP53* mutation in these samples, we quantified the levels of *TP53* mutation expression (RNA-mutated allele frequency, RMAF). Corresponding with Xi-large deletion, we found significantly higher *TP53* RMAF levels, indicative of *TP53* loss of heterozygosity in these samples (LOH; p<0.01; Wilcoxon test; Figure 2c). *TP53* mutation with a loss of the remaining *TP53* allele is reported to be the most typical pathogenic configuration [26]. These findings tie together *TP53* mutation, reduced *XIST* expression, and Xi-large deletions in aggressive BRCAs.

Beyond Xi deletions, amplifications of large regions of X chromosomes have been associated with several diseases and syndromes [16,37,38], and the duplication of the Xa has been found in BRCA cell lines [39]. We selected TCGA-BRCA samples from the Xi-unaltered group (710 tumour samples; Appendix A) and performed hierarchical clustering by the X-chromosome allele-specific copy number of the Xa. We identified two groups: a set of tumours with complete or extended Xa amplifications (Xa-large-amplification), and another set with a mostly unaltered Xa copy number (Xa-unaltered) (Figure 3a; Appendix A). Outstandingly, the Xa-large-amplification samples were most frequent among the most aggressive subtypes—Basal (62%; 46/74), Her2 (65%; 34/52), and LumB (69%; 91/131)—compared to LumA samples (37%; 171/453), (Figure 3b). Resembling patients with large Xi deletions, Mt-*TP53* samples were overrepresented in the Xa-large-amplification group (Mt-*TP53*: 68%; 121/178; Wt-*TP53*: 41%; 221/532) (Figure 3c). In contrast to the Xi-large-deletion samples, however, there was no significant alteration in *XIST* levels and X promoter methylation in the Xa-large-amplification group (Appendix A). The failure to detect either elevated *XIST* levels or higher methylation in association with Xa amplifications is consistent with dysregulation of dosage compensation by XCI machinery. This scenario would be anticipated to correspond with higher X-linked gene expression, as we will go on to explore.

Using our methodology to quantify RMAFs [28], we observe an association between Xa-large amplifications and elevated *TP53* RMAF levels (p<0.01; Wilcoxon test; Appendix A). Together, these findings highlight the synergy of WT-*TP53* with the integrity of the X chromosome, and also implicate the role of X-chromosome alterations in breast cancer aggressiveness, in conjunction with *TP53* mutation.

### 3.2. Breast Carcinomas with Large Xi Deletions Show Higher Incidence of Fully Expressed Mutations

We previously showed that males are at particular risk of compromising the integrity of the X chromosome’s p53 network, contributing to cancer sex-disparity [28]. Somatic mutations in the single male-X have an obligate expression, in contrast to those in females, where the XCI of a damaged second X-chromosome copy may maintain this network functionally intact. Specifically, it was shown that six mutated tumour-suppressor genes on the Xa are functionally compensated by their intact counterparts on the second allele, the Xi, as they escape XCI [11]. Importantly, we identified large deletions of the Xi that nullify protective diploidy and result in the full mRNA expression of mutations existing in the Xa [28]. Using RMAF levels, we classified somatic mutations in the X chromosome as: “expressed”, ‘’non-expressed”, and “escape or bi-allelic”. Noticeably, samples with large Xi deletions showed almost double the proportion of expressed somatic mutations in the X chromosome (p<0.001; ChiSq test; Appendix A). These findings reinforce the association between large Xi deletions and more aggressive tumours, as there is a higher probability of a cancer hallmark event—oncogene activation and tumour-suppressor inactivation—via somatic mutations.

### 3.3. Gene Signatures Associated with Large Xi Deletions and Large Xa Amplifications Show Correlations with Poor BRCA Survival

In order to further understand the impact of large Xi deletions, we performed a differential expression analysis between the Xi-large-deletion group (305 tumours) and a selected group of samples with no large copy number aberrations and unaltered levels of X promoter methylation (201 selected tumours) (Appendix A). Undertaking differential expression analysis without adjusting for PAM50 subtypes could bias our findings, given the uneven number of subtype samples, reflecting only the biological differences between the overrepresented subtypes in each group. Consequently, the analysis was performed in a BRCA PAM50 subtype-aware manner: by including the subtype as a covariate. We focused on downregulated genes in the Xi-large-deletion group in order to identify the loss of gene expression as a result of Xi deletions. In the X chromosome, there were 63 genes with significant downregulation (FDR≤0.05, logFC<0; Appendix A). We found three tumour suppressors identified in the COSMIC database [40] to be significantly downregulated in the Xi-large-deletion samples. Of note, the tumour suppressors KDM5C and KDM6A have been previously identified to play fundamental roles in cancer sex-disparity, as they escape XCI to retain their function when mutated [11], while ZRSR2 mutations/loss have been associated with sex-disparity in blood cancers [41]. Importantly, large deletions of Xi void the possibility of extra protection from a non-mutated second allele, relevant to the more aggressive breast cancer subtypes. This is highlighted by an overrepresentation of downregulated genes in the pseudoautosomal region 1 (PAR1, *p* < 0.001; hypergeometric test), a genomic region that sustains consistent XCI escaping [4]. The function of additional downregulated genes is discussed below, in the context of promoter methylation. Furthermore, we used the 63 X-linked significantly downregulated genes (FDR≤0.05, logFC<0) and defined an Xi-deletion signature; we then assessed its prognostic significance in METABRIC and found it to be highly predictive of disease-specific survival (HR = 13, p≈0; HR test; Figure 4a), strengthening the association between large Xi deletions and BRCA aggressiveness.

Similarly, we set out to explore the functional impact of large Xa amplifications in BRCA and performed a differential expression analysis between the Xa-large-amplification group (342 tumours) and selected unaltered tumours (201 samples; Appendix A). We focused on genes upregulated in the Xa-large-amplification group to understand the functional advantages gained by the extra Xa copies, and found 225 X-linked genes showing significant differences between these two groups (FDR≤0.05, logFC>0; Appendix A).

Strikingly, among the upregulated genes in the Xa-large-amplification group, we identified genes of the XCU developmental process that are deployed to fine-tune the XCI and elevate the expression of the Xa, in both males and females [42]. Of the 66 genes linked to the XCU, we identified that 50% (33/66) of these genes were significantly elevated in these BRCA samples (Appendix A, Tab#2). In addition, genomic regions known to host XCI-escaping genes, PAR1 and PAR2, showed significant proportions of upregulated genes (*p* = 0.02 and *p* = 0.04, respectively; hypergeometric test). These findings predict that these normal regulatory processes during development (XCU and XCI) are being exploited for their advantage during BRCA development in this subset of samples.

Furthermore, 35 of the Xa-amplified upregulated genes have functional links to *TP53* (39% of the p53 STING set genes; 35/90; Appendix A, Tab#3) [28]. Among these are genes with firmly established links to cancer promotion [28]—for example, HUWE1, an important E3 ligase of p53 [43] that may also promote cancer independently [44], and HDAC6 and HDAC8, which are being actively explored as therapeutic targets [45].

Additional known oncogenes were identified among this group. Specific examples, ARAF1 and ELK1, have been previously identified as oncogenes in testicular cancers with X-chromosome amplifications [46], and GRPR has been associated with an increased risk of lung cancer in females [47]. We used the 225 genes as an Xa-amplification signature and found high expression levels to be significantly prognostic of poor BRCA patient survival in METABRIC (HR = 3.2, *p* < 0.001; HR test; Figure 4b).

### 3.4. Aberrant X-Chromosome Inactivation Is Associated with Breast Cancer Survival

In addition to Xi-large deletions or Xa-amplifications, we reasoned that our findings in Figure 1a,b indicated that aberrant XCI mechanisms could also contribute to the initiation and propagation of BRCAs. We chose to focus on abnormal X gene promoter methylation to identify aberrant XCI. We performed hierarchical clustering using the promoter methylation of X-chromosome genes. From the 368 TCGA-BRCA samples that had no large X copy number alterations (Xa-unaltered group; Appendix A), only 280 have available methylation data.

Among the 280 samples, our clustering highlighted three main X promoter methylation groups: high, low, and unaltered (54, 25, and 201 samples, respectively; Figure 5a), revealing highly variable X-chromosome mean promoter methylation (p≈0; K–W test; Figure 5b), with the unaltered group showing methylation levels similar to normal-matched samples (Appendix A). Of clinical relevance, samples exhibiting high methylation and no large CNA also showed an increase in fully expressed mutations relative to their counterparts with low methylation levels (Appendix A). This is in concordance with the excessive XCI enacted through promoter methylation, limiting the capacity of Xi genes to escape and compensate for the mutated Xa.

Exploring the selective advantage gained by X-chromosome CNAs, we accessed the BRCA-TCGA assay for transposose-accessible chromatin with sequencing (ATAC-seq) peak calls [30]. Open chromatin regions are special regions of the human genome that can be accessed by DNA regulatory elements such as DNA methylation [48]. The BRCA samples adopted for this analysis were not altered for Xi, Xa, or X promoter methylation levels (Selected Tumours; Appendix A). We compared the gene promoter methylation levels between genes located inside or outside open chromatin regions, according to the gene signatures we defined for Xi-deletions and Xa-amplifications (Appendix A). We found that genes in the Xi-deletion signature, located in open chromatin areas, showed significantly lower promoter methylation than signature genes outside open chromatin (p=0.01; *t*-test). Consequently, these lowly methylated genes in open chromatin are likely to be prone to escape XCI. Our data suggest that genomic deletion is an effective mechanism to downregulate the expression of these genes during cancer progression. In contrast, genes in the Xa-amplification signature showed no significant differences in promoter methylation between open or closed chromatin areas, signalling no consistent XCI escaping and implicating genomic amplification as a mechanism to achieve additional expression.

In order to build a functional signature representative of high XCI levels, we performed gene differential expression analysis between samples in the high and unaltered methylation groups, with a focus exclusively on downregulated genes (FDR≤0.05, logFC<0; Appendix A). We found that higher levels of this signature significantly correlated with worse disease survival in METABRIC samples (p=0.008; HR test; Figure 6a). These results indicate that BRCA samples with abnormally high levels of promoter methylation do not benefit from XCI escaping and additional protection, as is consistent with previous studies [11]. Similarly, we compared samples between the low and unaltered methylation groups, to capture the effects of low or dysregulated XCI, focusing on upregulated genes (FDR≤0.05, logFC>0; Appendix A). The low-methylation signature is also associated with worse prognosis in METABRIC (p=0.058; HR test; Figure 6b).

### 3.5. Aggressive BRCA Phenotypes Are Accomplished via Different Mechanisms

Speculating that silencing the expression of X genes, such as those encoding tumour suppressors, is likely to offer a selective growth advantage to BRCAs, we reasoned that common outcomes could be achieved by either Xi deletion or promoter methylation. Aligning with this reasoning, we identified 27 genes in common between the Xi-deletion genes (27/63; 43%) and the high-methylation group (27/80; 34%; Appendix A, Tab#1).

Undertaking gene ontology analysis exclusively on candidates from a single chromosome, such as the X chromosome, offers very little statistical power, as pathways typically comprise gene products from across the genome. As an alternative, we searched the literature for evidence that the signature genes were involved in cancer. This approach identified more than half the genes (17/27) in these BRCA samples that have recently been recognised for their tumour-suppressive capacities in a range of sporadic cancers, with others apparently underexplored (Appendix A, Tab#2). Several prime examples are functionally linked to metabolism: monoamine oxidase A (MAOA) inhibits aerobic glycolysis and immunity in lung cancer [49]; Integral Membrane Protein 2A (ITM2A) in BRCA induces PD-1 and higher tumour infiltrating lymphocyte (TIL) infiltration [50]; and GRB2-associated binding protein 3 (GAB3) drives natural killer cell priming and expansion [51].

These findings confirm our methodology for identifying genes of biological relevance and further support the concept that these specific X-linked genes are not randomly downregulated, but, rather, are depleted under selective pressure to provide a growth advantage to the tumour.

We also postulated that large Xa-amplifications are likely to be selected to offer a growth advantage, and low-promoter methylation may duplicate this outcome. While only five genes were identified with significantly reduced methylation, 80% (4/5) overlapped with the Xa-amplified genes (Appendix A, Tab#2), and their capabilities to promote cancer is extensively documented. Notably, Gamma-Aminobutyric Acid Type A Receptor (GABRA3) amplification promotes BRCA metastasis [52]. As their name suggests, the melanoma-associated antigen (MAGE) family has been linked to cancers [53], including the following members: A3 (an inhibitor of p53 signalling) [54]; A6 (with both A3 and A6 expression linked to BRCA) [55]; and C2 (reported to promote the epithelial–mesenchymal transition in BRCA, and an activator of p53 ubiquitination) [43,56,57]. Interestingly, *MAGEA4* has been associated with XCI dysregulation in ovarian cancers [13]. Overall, these previously unreported clinical associations between Xi and Xa aberrations, cancer aggressiveness, and survival outcome offer new insight into the understudied complexities of the X chromosome in breast cancer and links to *TP53* alterations.

## 4. Discussion

The tumour suppressor p53 is acclaimed for its critical protection against genomic DNA damage generally, with new aspects of its activities continuing to emerge [23,58,59]. Astonishingly, little is known, however, regarding its engagement with the sex chromosomes, the allosomes. Further exploration of the X chromosome and p53 was prompted by our work defining sex-distinct features: in a p53-X gene network affected in cancers [28], and in the p53 activation of *XIST* expression during development [27]. In male cancers, we uncovered peculiar vulnerabilities associated with gene mutations on a haploid X chromosome [28]. Two X chromosomes in females, by contrast, offer more robust protection [11]; however, despite this inherent advantage, some females do develop cancer in diverse forms of aggressiveness, which led us to question the role of the X chromosome’s integrity in female cancer progression. We chose to study this phenomenon in BRCA, as it is the most prevalent cancer diagnosed in females and offers the largest data cohorts.

We uncovered novel links between WT*-TP53* and consistent XCI. Firstly, the occurrence of WT*-TP53* aligns with the proper control of two prime elements of XCI: X gene promoter methylation and *XIST* expression. In contrast, both these activities are depleted when *TP53* is mutated (Figure 1). This insinuates an original role for WT-p53 in adult XCI maintenance, beyond its function in development [27]. Secondly, as cancer of the breast is the most frequently diagnosed cancer type in females, breasts are either more exposed to carcinogens or are less capable of repairing damaged DNA, or even both, suggesting p53 and XCI encounter tissue-specific risks. In addition, as breast cancers are rare in males, the hormonal context is presumed to influence this interaction and implies a sex-specific nature of p53 in this tumour-suppressor activity.

DNA methylation homeostasis has been linked to WT-p53 in embryonic stem cells [60]. In BrCa, *TP53* mutation has been linked to altered methylation, associated with epigenomic instability and correlated with tumour grade and stage [2]. A novelty of our study is to expose the significant correlation between *WT-p53* and proper X chromosome promoter methylation (Figure 1c) and to demonstrate the survival consequences that ensue when that control is broken (Figure 1e).

An indication that proper *XIST* expression correlates with WT-*TP53* in female adult breasts draws questions as to whether *TP53* status is also relevant to non-reproductive female adult cancers. This is relevant as to whether *XIST* function appears crucial, notably in specific blood and intestinal cancers [61,62]. It is also of interest to ask whether p53 is acting properly in immune contexts where *XIST* dysfunction is noted, including autoimmune disease and COVID-19 infection [5]. The emerging role of p53 in controlling immunity, including in a cancer context [63], provides a rational basis for exploring such connections.

Together, our original findings indicate a new role for WT-p53 as a fundamental guardian of the female X chromosome. This involves the proper regulation of X-chromosome expression across the full female lifespan, from early embryogenesis [27] through adulthood. The mutation of *TP53* or reduced protein levels of WT-p53, either in response to the altered expression or elevated activities of its negative regulators, appears as a serious cancer liability for females and, most particularly, for BRCA. Our earlier calculation that females are at a lower risk of developing *TP53* mutations in non-reproductive cancers than males [28] now also appears vitally relevant for female breasts, where the consequences of early p53 malfunction would have devastating consequences for the sustenance of mammalian offspring and, in turn, negatively affect reproductive fecundity. In this light, it is particularly relevant that BRCA is also the most frequent cancer type among adult women with inherited *TP53* mutations, from Li–Fraumeni syndrome (LFS) families [64]. Our study directs the application of our analytical methods to the role of the X chromosome in such families, with attention to X copy number (CN) considerations, to avoid the masking of differences (e.g., as pertinent to an earlier, small study of 10 BRCA LFS patients) [65]. We predict that our approach will offer prognostic relevance associated with specific types of *TP53* mutations in LFS BRCA. In the context of female cancers and, particularly, BRCA, we are not able to ascertain from our data whether the loss of *TP53* is equivalent to its mutation. We suggest that it will be highly relevant to assess in future studies whether mutant p53 exhibits a GOF in XCI, both for somatic cancers and also for LFS patients.

Evidence of the disruption of XCI mechanisms linked to *TP53* mutation (Figure 1) primed our in-depth study of X-chromosome fidelity in the BRCA cohorts. Exploring for ramifications of dysregulated XCI led us to identify the significant enrichment for *TP53* mutation among tumours with large Xi deletions (Figure 2) and large Xa amplifications (Figure 3). Our findings infer a significant role for WT-p53, not only as a suppressor of altered X-chromosome CN but also of major X deletion and duplication events. These original insights regarding WT-p53 and the maintenance of proper X-chromosome ploidy in adult female breast tissues are in keeping with the known role of WT-p53 in suppressing aneuploidy generally [24,66].

A particularly aggressive cancer progression is associated with mutant *TP53* RNA that has undergone LOH [26]. In alignment, we found that BRCA patients with the poorest survival exhibited LOH of mutant *TP53,* coincident with a significant enrichment for large Xi deletions and Xa amplifications (Appendix A, respectively, where *TP53* RMAF approaches 1.0). Consistent with poor survival among Basal subtypes, *TP53* mutations and X-chromosome aberrations were enriched among this tumour type. Notably, however, our methodology predicts that other genetic factors beyond subtyping are also prognostic and of value to explore, for understanding disease progression and potentially for identifying advanced treatment opportunities.

The functional consequences of deletions and amplifications in the form of differentially expressed genes were not uniform across the X chromosome. The identified functions of the genes that were deleted and those amplified suggest that their manipulation would offer an advantage to the growing tumours—most notably, the deletion of TS genes (e.g., immunity ITM2A, and metabolism genes; Appendix A, Tab#2) and the amplification of oncogenes (e.g., MAGE genes; Appendix A, Tab#3). The reiteration of altered gene expression by either increased (for TS genes) or decreased (for oncogenes) promoter methylation reinforces the relevance of these genes to cancer. These findings imply that selective pressure is driving these X and XCI deregulation processes. In addition, as we highlighted, PAR regions are particularly affected. Further exploration of affected genomic regions and the identification of a regional map of XCI escaping could offer diagnostic advantages and insight into cancer aggressiveness.

Our findings provide a mechanistic insight into previously observed correlations between poor BRCA patient survival and mutant p53 status [67]. Our novel approach has generated molecular signatures that are predictive of survival (Figure 4 and Figure 6). Our novel signatures comprise genes that are of value to explore for therapeutic potential (e.g., HDAC genes; Appendix A, Tab#3; [45]).

The genes comprising the BRCA signatures that we identified validate our bioinformatic approach. Notably, we provide novel insight into genetic aberrations in BRCAs beyond the criteria adopted for standard subtype classifications. The clinical application and therapeutic value of these findings are waiting to be explored.

Our methodology can now be applied to other cancers in females. We anticipate that contextual tumour environmental pressures will influence the nature of the X-chromosome aberrations. The reason why tissue-type proclivity exists and BRCAs are the most prevalent cancer in females was not explored in this study; however, we note the positive loop between estrogen, p53, and the major p53 negative regulator, MDM2 [68]. Aligning with hormonal influence, particularly, X-chromosome dysregulation is rife in ovarian cancer, another prominent reproductive cancer [12]. Around 96% of high-grade serous ovarian cancers carry *TP53* mutations [69], and our results support the rationale for exploring a link between the two in ovarian cancers.

Beyond female cancers, as a number of genes in our BRCA signatures had previously emerged as cancer genes in males—for example, RGN, a calcium regulator found to be a TS in prostate cancer (PrCa), and COL4A6, whose downregulation and hypermethylation is linked to progression and metastasis of PrCa (Appendix A)—we suggest that further investigation of our listed genes is warranted, in cancers of both reproductive and non-reproductive organs of males and females.

For the first time, using NGS, we have comprehensively characterised the role of the X chromosome in BRCA aggressiveness. In summary, our data strongly imply that X-chromosome aberrations and disruptions of elemental X-chromosome biological processes dictate BRCA development and patient outcome. Remarkably, our methodology can accurately discriminate BRCA patient prognoses based solely on the analysis of the X chromosome. With tight links to *TP53*, our findings add relevant molecular understanding to BRCA, beyond the standardly used BRCA molecular–pathology subtyping. These original insights into the fundamental role of X chromosomes in BRCA defence that our findings unlock argue a strong case for a more thorough clinical examination of X-chromosome aberrations in the analysis and treatment of BRCA.

## 5. Conclusions

The existence of a single X chromosome in males has been reported as a peculiar vulnerability that poses high cancer risks, yet females do develop cancers, albeit at different rates. The intricacies of XCI have posed a technical barrier to X chromosome analyses in females. By applying our novel analytical method to a breadth of female BRCA samples, we have uncovered vital new links between BRCA aggressiveness, the integrity of the X chromosome and the major tumour suppressor *TP53*. Our original approach unlocks exciting new opportunities to study the spectrum of female cancers.

## Figures and Tables

**Figure 1 cells-12-02245-f001:**
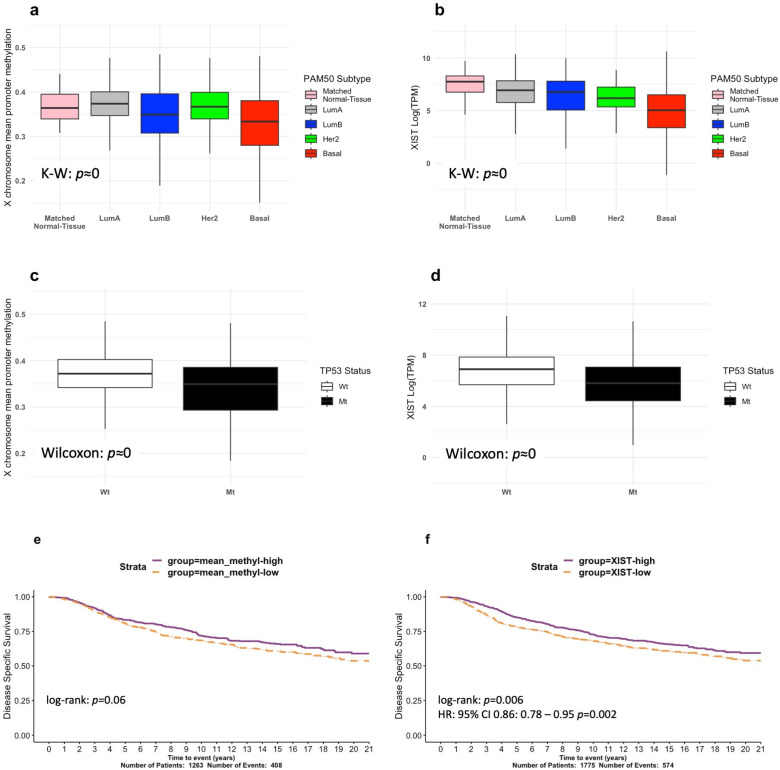
Exploring X-chromosome methylation and XIST expression in the TCGA-BRCA and METABRIC cohorts. (**a**) TCGA-BRCA X -chromosome mean promoter methylation stratified by PAM50 subtype. (**b**) TCGA-BRCA XIST expression stratified by PAM50 subtype. (**c**) TCGA-BRCA X-chromosome mean methylation levels stratified by TP53 mutation status. (**d**) TCGA-BRCA XIST expression stratified by TP53 mutation status. METABRIC Kaplan–Meier survival curves for disease-free survival showing prognostic separation and log-rank test *p*-values for (**e**) mean X promoter methylation and (**f**) XIST expression.

**Figure 2 cells-12-02245-f002:**
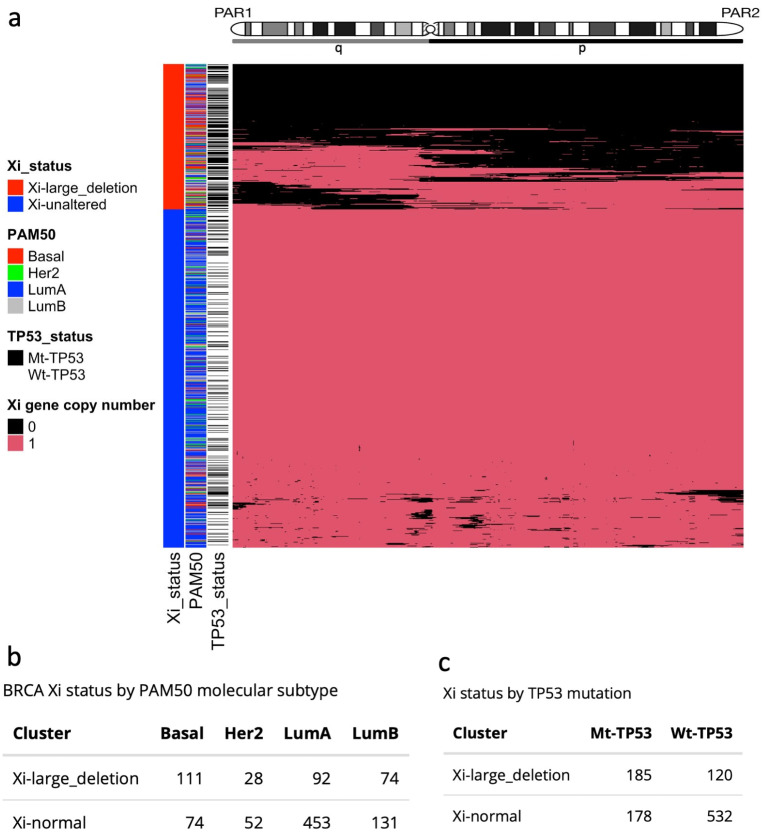
Large Xi deletions in the TCGA-BRCA cohort. (**a**) Heatmap and hierarchical clustering of Xi deletions define two groups: Xi-large-deletions and Xi-unaltered. (**b**) Number of BRCA molecular subtypes by Xi aberration groups. (**c**) TP53 mutation status by Xi aberration group.

**Figure 3 cells-12-02245-f003:**
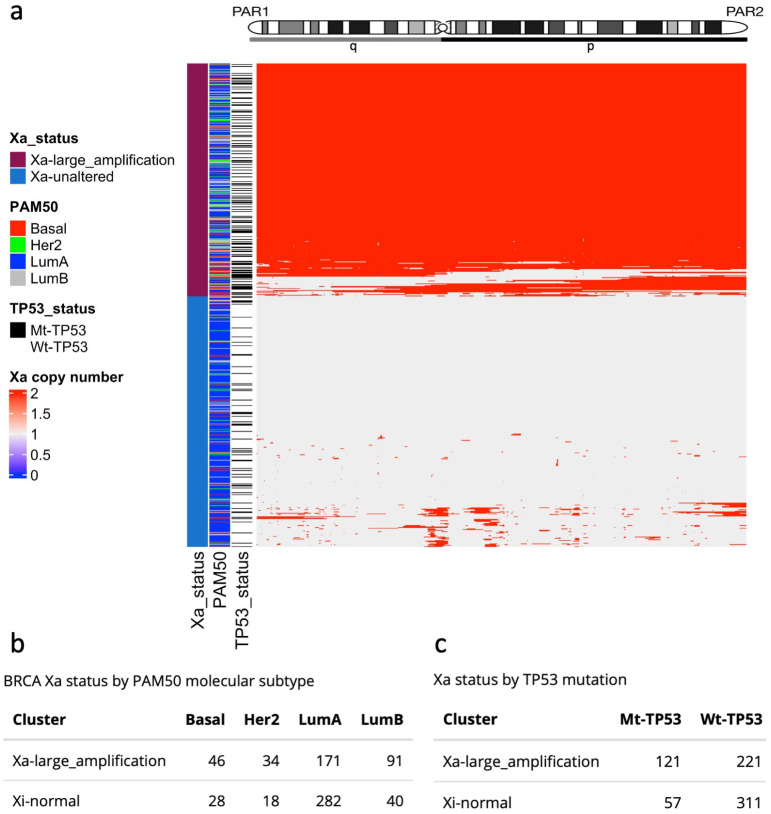
Large Xa amplifications in the TCGA-BRCA cohort. (**a**) Heatmap and hierarchical clustering of copy number Xa amplifications define two groups: Xa-large-amplifications and Xa-unaltered. (**b**) Number of BRCA molecular subtypes by Xa amplification group. (**c**) TP53 mutation status by Xa amplification group.

**Figure 4 cells-12-02245-f004:**
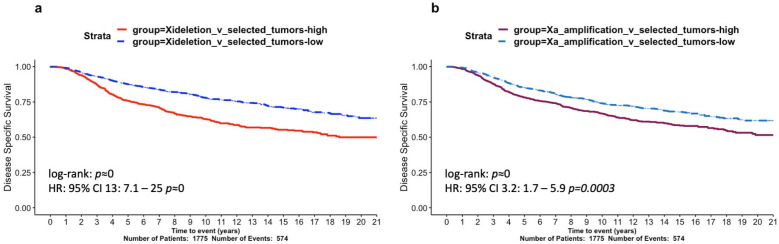
Survival analysis of X-chromosome aberration signatures in the METABRIC cohort. Kaplan–Meier curves of 20-year disease-specific survival assessing two gene signatures: (**a**) Xi-deletion and (**b**) Xa-amplification. Hazard ratios derived from Cox regression models were derived and *p*-values are shown.

**Figure 5 cells-12-02245-f005:**
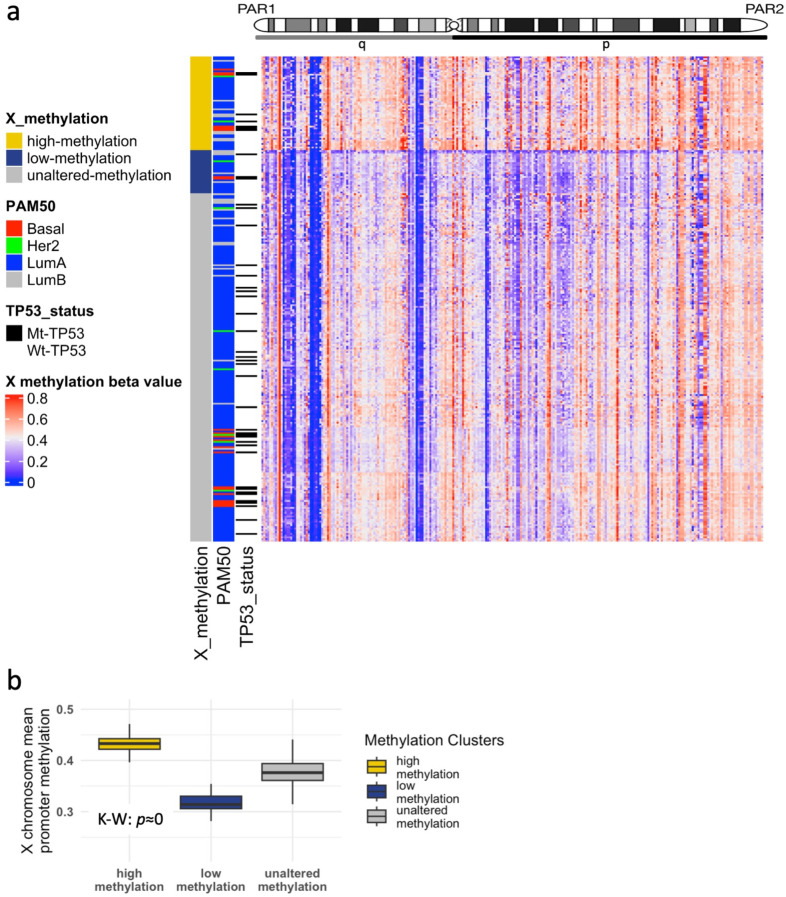
Abnormal X-chromosome methylation in the TCGA-BRCA cohort. (**a**) Heatmap and hierarchical clustering of methylation beta-values defines three groups: high-methylation, low-methylation, and unaltered-methylation. (**b**) Mean X promoter methylation beta-values comparing the three groups.

**Figure 6 cells-12-02245-f006:**
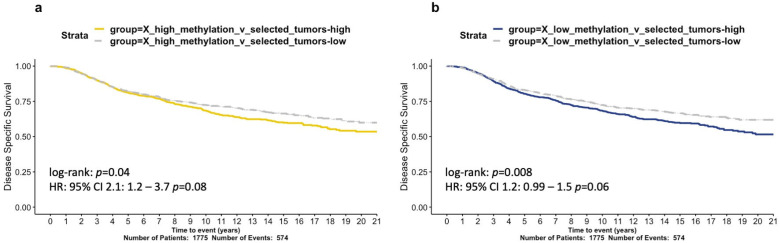
Survival analysis of abnormal X chromosome methylation signatures in the METABRIC cohort. Kaplan-Meier curves of 20-year disease specific survival assessing two gene signatures. (**a**) X-high-methylation and (**b**) X-low-methylation. Hazard ratios derived from Cox regression models were derived and *p*-values are shown.

## Data Availability

The TCGA data can be accessed in https://portal.gdc.cancer.gov. The METABRIC project and publicly available data can be accessed in https://www.cbioportal.org.

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
