# Peer review of "Establishing the Link between X-Chromosome Aberrations and TP53 Status, with Breast Cancer Patient Outcomes"

_cells, 2023, doi:10.3390/cells12182245_

Round 1

Reviewer 1 Report

This paper revealed the relationship between X chromosome aberrations and breast cancer outcome. This study was built up upon the 2019 Nature Communications paper published by the same team discussing how cancer sex-disparity is affected by the major tumor suppressor TP53 and its X chromosome network. Understanding breast cancer at a molecular level is highly important to its diagnostics, prevention and treatment. This paper for the first time related breast cancer aggressiveness with patient's X chromosome genetic fingerprint with bioinformatics tools, which is highly novel and significant. This paper is well written, with sound discussions/conclusions supported by ample data. I do not have additional suggestions to make and thus suggest publish as is. 

Author Response

We thank Reviewer 1 for their favourable comments and will be delighted if our research leads to new insights entering the clinic.

Reviewer 2 Report

The authors choose to focus their science on chromosome X, particularly the inactivated form, in breast cancer. The premise alone is important, as sex-specific differences in cancer are poorly understood, and other published chromosome alteration studies often ignore sex chromosomes for technical reasons. While the study is descriptive and reanalyzes existing data, the authors create useful figures for rapid comprehension of the chromosome X data. This is not trivial; each analysis no doubt required professional coding and quality control to yield these nice figures. In particular, the difference between amplified Xa and losses in Xi are interesting, as if it were truly random there should be no difference.

One minor concern is the Kaplan-Meier data presentation. Cancers with copy-number alterations usually perform worse, as do basal breast cancers (and basal breast cancers have more chromosome alterations). The authors need to also provide well-controlled analyses (perhaps in the supplement) in which only basal cancers of similar overall aneuploidy are compared. This will inform how special chromosome X is, if at all, in terms of chemotherapy prognosis.

There are some minor typos - the authors should revisit their manuscript in full prior to publication.

Nice study. Thank you for sharing with other chromosome biologists.

Author Response

We thank Reviewer 3 for their positive comments. In regard of their concern, we contemplated a similar approach to validate the prognostic power of our gene signatures, but we acknowledge the limitations of the available data. Unfortunately, the number of basal samples present in TCGA does not allow the performance of survival analysis, after segmenting by aneuploidy groups. For example, from the 190 Basal samples in TCGA, only 13 have no major copy number changes in the X chromosome (Xa amplifications, Xi deletions). However, we took the 201 unaltered X breast cancers from TCGA (the selected group in our manuscript) and found our Xi-deletion signature to be prognostic over a 5-year survival period using a log-rank test (p-value = 0.03), demonstrating the value of the signature in a diploid X context. We chose not to show this data due to the low number of events (10). In addition, our differential expression strategy integrates molecular subtypes as a confounding factor to minimise the effect of subtype biology in our signatures. We hope that large initiatives like TCGA will allow for more detailed analyses in the future.

We trust that our explanation of the technical limitations of Reviewer 3’s request meets their concern.

We will correct the typos accordingly.
